# The Effects of Fecal Microbial Transplantation on the Symptoms in Autism Spectrum Disorder, Gut Microbiota and Metabolites: A Scoping Review

**DOI:** 10.3390/microorganisms13061290

**Published:** 2025-05-31

**Authors:** Ignazio Maniscalco, Piotr Bartochowski, Vittoria Priori, Sidonia Paula Iancau, Michele De Francesco, Marco Innamorati, Natalia Jagodzinska, Giancarlo Giupponi, Luca Masucci, Andreas Conca, Magdalena Mroczek

**Affiliations:** 1Department of Psychiatry, Hospital of Bolzano (SABES-ASDAA), Teaching Hospital of Paracelsus Medical, 39100 Bolzano, Italy; ignazio.maniscalco@sabes.it (I.M.); vittoria.priori@sabes.it (V.P.); michele.defrancesco@sabes.it (M.D.F.); giancarlo.giupponi@sabes.it (G.G.); andreas.conca@sabes.it (A.C.); 2BC2M, Faculty of Pharmacy, University of Montpellier, 34090 Montpellier, France; piotr.bartochowski1@gmail.com; 3Residence School in Psychiatry, Faculty of Medicine and Psychology, Sant’Andrea Hospital, Sapienza University, 00185 Rome, Italy; sidoniapiancau@gmail.com; 4Azienda Ospedaliera San Camillo Forlanini, Cir.ne Gianicolense 87, 00152 Roma, Italy; 5Department of Health and Life Sciences, European University of Rome, 00163 Roma, Italy; marco.innamorati@unier.it; 6School of Clinical Medicine, University of Cambridge, Cambridge CB2 1TN, UK; nmj35@cam.ac.uk; 7Microbiology, Fondazione Policlinico Universitario Agostino Gemelli IRCCS, Catholic University of Medicine, 00168 Roma, Italy; luca.masucci@policlinicogemelli.it; 8Department of Consultation-Liaison Psychiatry and Psychosomatic Medicine, University Hospital Zurich, University of Zurich, 8006 Zurich, Switzerland

**Keywords:** microbiota, FMT, fecal microbial transplantation, ASD, autism spectrum disorders

## Abstract

The bilateral interaction between the brain and the gut has recently been on the spectrum of researchers’ interests, including complex neural, endocrinological, and immunological signaling pathways. The first case reports and clinical studies have already reported that delivering microbes through fecal microbial transplantation (FMT) may alleviate symptoms of psychiatric disorders. Therefore, modifying the gut microbiota through FMT holds promise as a potential treatment for psychiatric diseases. This scoping review assessed studies from PubMed related to FMT in autism spectrum disorder and attention deficit hyperactivity disorder. The evaluation included nine clinical studies and case reports. The beneficial and persistent effect on the autism spectrum disorder (ASD) symptoms has been reported. Also, an increased microflora diversity and altered levels of neurometabolites in serum were identified, albeit with a tendency to return to baseline over time. The microbiome–gut–brain axis could provide new targets for preventing and treating psychiatric disorders. However, a recent large randomized clinical trial has shed light on the previously collected data and suggested a possible contribution of the placebo effect. This highlights the necessity of large randomized double-blind studies to reliably assess the effect of FMT in ASD.

## 1. Introduction

The term gut microbiota describes a complex ecosystem of various microorganisms inhabiting the intestinal lumen. The gut microbiota is composed of bacteria, eukaryotes, and archaea [1]. These organisms remain in a mutualistic relationship with the host, in which multiple metabolic pathways are interconnected. Among these domains, bacteria predominate, and their contribution to the gut microbiota–host relationship has been best understood [2]. Gut bacteria play a key role in a variety of processes, including nutrient metabolism, short-chain fatty acids (SCFAs) production, vitamin synthesis, immunomodulation, antimicrobial protection, and maintenance of the intestinal barrier integrity [2].

While, in health, the relationship between the host and the gut microbiota is symbiotic, in chronic illness, adverse changes in the composition and metabolism of the gut microbiota, called dysbiosis, are observed, which may contribute to disease progression. Dysbiosis has also been associated with several psychiatric diseases, including anxiety and autism spectrum disorder (ASD) [3].

Autism spectrum disorders are neurodevelopmental conditions characterized by early difficulties in social communication and repetitive behaviors as primary symptoms [4]. ASD affects approximately 1% of the global population and is more common in males than females, with a male-to-female ratio of approximately 4:1 [5]. Individuals with autism often experience challenges in social cognition, executive function, and sensory processing. Several co-occurring conditions have been identified, with intellectual disability, language disorders, and attention deficit hyperactivity disorder among the most commonly reported. The underlying causes of ASD are thought to be multifactorial, with several hypotheses proposed, including anatomical, genetic, and environmental factors. Prenatal and perinatal risk factors have also been highlighted, but due to the heterogeneity of the findings, no single factor has been definitively linked to ASD [6]. An important puzzle in the development of ASD may lie in a disrupted relationship along the brain–gut axis. The concept that the intestinal microflora can affect the brain, including higher cognitive functions, is well established and supported by animal studies [7,8,9]. The gut can influence the brain through a variety of mechanisms, including gut microbiota composition, immunological processes, production of neuroactive metabolites, and vagal interaction [8]. Recent years has seen the development of greater understanding of the brain–gut axis, with the central nervous, gastrointestinal, and immune systems bilateral interactions acting as one organ [10,11]. Not only can the intestinal microbiota influence the brain, but the opposite can occur, through the brain to the enteric signaling, both directly (through regulating mobility, secretion, and gut permeability), and indirectly (through secretion of the active substances through the *lamina propria*) [10,12]. Interestingly, the influence of the gut microbiota on the nervous system can already be observed during prenatal life; however, these are interactions between the maternal gut microbiota and the developing nervous system of the fetus. Adverse factors, such as infections, poor diet, obesity, and medication use, may lead to gut-related impairment of the immune environment, shaping both the neurodevelopment (see [13], and, in preclinical studies, [14]) and the early formation of the fetal immune system [15]. However, it remains uncertain whether maternal gut microbes can cross the placenta and colonize the fetus [16]. For this reason, it is generally accepted that the initial inoculation of the infant’s microbiome occurs at birth. During vaginal delivery, the newborn is inoculated by the maternal vaginal microbiota; whereas, in the case of C-section, the colonizing microbes originate from maternal skin and the hospital environment. Numerous studies have indicated that the microbiota composition associated with C-sections is more pro-inflammatory, which may negatively impact the proper development of both the immune and nervous systems [17]. A C-section birth is recognized as a risk factor for ASD [18], and animal studies have shown that C-section delivery increases the likelihood of anxiety-like behavior and behavioral deficits in offspring [19]. Later in life, individuals with ASD show an increased abundance of taxa from the *Firmicutes* and *Pseudomonadota* phyla, while the prevalence of *Bacteroidetes* is reduced compared to neurotypical controls [20]. This shift may be influenced by multiple factors, including poor dietary habits shaped by the sensory hypersensitivity commonly observed in ASD [21,22], host genetics [23], medication use, and the course of the disorder itself. Interestingly, gastrointestinal (GI) problems are also frequent in individuals with ASD, affecting nearly 91% of them [24]. These GI issues include chronic constipation, abdominal pain, diarrhea, reflux, inflammatory bowel disease, celiac disease, and Crohn’s disease [25], suggesting a possible underlying chronic inflammatory state in the intestine. Bacterial dysbiosis has been associated with autism, although its causative role remains unclear.

Fecal microbiota transplantation (FMT) is the transfer of stool from a healthy donor to a recipient to restore gut microbiome diversity and balance. Although the first protocol providing the minimum general steps for FMT [26], further studies to better determine terms of route, protocols optimization, donor-recipient pairing and donor optimization and duration of therapy [27,28]. Both frozen and fresh protocols proved to be successful [26]. FMT can be delivered in multiple routes, among them by nasogastric/nasojejunal tube, by endoscopy, by oral capsules, and by the lower gastrointestinal route (LGI), like retention enema, sigmoidoscopy, or colonoscopy. Capsule delivery, either traditional or colon-targeted, is the most commonly applied administration route [29]. Recent American Gastroenterological Association (AGA) guidelines have recommended FMT to prevent recurrent *C. difficile* in select patients, but have suggested against the clinical use of FMT for inflammatory bowel diseases or irritable bowel syndrome [27] and, at the same time, have recommended the European consensus on FMT, which has also advised the use of FMT for the treatment of *C. difficile* in specialized centers, but has found no strong evidence for other applications [26]. Although potential use in other indications, such as other gastrointestinal, metabolic, and neurological disorders has been suggested [30]. FMT shows several advantages over probiotic supplementation in ASD: it has longer persistence than probiotics [31] and is more feasible, considering the food selectivity of ASD children. Additionally, alternative treatment options for ASD are limited.

With growing knowledge about the role of gut microbiota in ASD children, there is increasing interest in microbiota-targeted therapies as an alternative or adjunct to traditional treatment approaches. In this review article, we aim to summarize the current clinical findings on the use of FMT in ASD treatment and their pitfalls, and to discuss the potential mechanisms underlying its effects.

## 2. Article Search

To investigate the effect of FMT on ASD, a systematic search was conducted in the PubMed database for relevant published materials. Additionally, references of the manuscripts screened were searched for suitable publications fulfilling the inclusion criteria. The search strategies were developed by one author (I.M.) and further refined through group discussion The following search query was used: “fecal microbiota transplantation” OR “FMT” OR “fecal transplant” OR “microbiota transfer therapy” OR “washed microbiota transplantation” AND “autism” OR “autism spectrum disorder” OR “ASD” OR “Asperger syndrome” OR “attention deficit hyperactivity disorder” OR “ADHD” AND “gastrointestinal symptoms” OR “gut microbiota” OR “microbiome” OR “clinical improvement” OR “treatment” OR “therapy” OR “intervention” OR “case report” OR “pediatrics” OR “children” OR “adolescents” OR “repeated transplantation” OR “washed microbiota” OR “intestinal microbiota” OR “fecal therapy” OR “stool transfer”. The search was performed on 15 March 2025.

## 3. Methods

The methodology aimed to be in line with the PRISMA Extension for Scoping Reviews (PRISMA-ScR) [32] and, if any divergence appeared, they were discussed in the limitations section of the discussion. Three researchers (I.M., V.P., and M.M.) independently screened the titles and abstracts of the records and discussed any disagreements until a consensus was reached. A total of 148 hits were identified. The inclusion criteria included the following: (1) studies must not be older than 10 years (*n* = 144); (2) the full text must be available (*n* = 143); (3) they must be original studies (*n* = 67); (4) they must be articles reporting clinical outcomes after FMT on humans in ASD and described a measure of treatment effectiveness, for example, by including single measures of validated psychometric tests, and/or by incorporating one or two dimensions of such measures (see PRISMA workflow; Figure 1). Quantitative and qualitative studies were included (*n* = 9). The exclusion criteria included (1) reviews and preprints, (2) preclinical studies, (3) not fully available articles, and (4) articles not written in English. Next, two researchers (I.M. and V.P.) independently screened full-text articles for inclusion. In the case of disagreement, a consensus was reached on inclusion or exclusion by discussion and, if necessary, the third researcher (M.M.) was consulted. Studies assessing clinical outcomes and measures have been included. Bacterial species analysis or neurotransmitters and metabolites analyses in sera were not required. Nine studies, including six clinical trials and three case reports, were selected. The strength of the study evidence was assessed using the GRADE approach [33]. No registration was performed for this scoping review, and therefore no registration number was assigned. The PRISMA workflow was developed with PRISMA2020: An R package (v1.1.1.)and Shiny app (2022) [34].

## 4. Impact of FMT on ASD Symptoms

During the literature search, nine publications were selected for analysis. To evaluate the treatment results several different scales were applied (for the description see the Appendix A) with CARS (*n* = 7), ABC (*n* = 6), and SRS (*n* = 4) most consistently reported (Table 1).

One of the first studies of FMT on 18 young patients (aged 7–16 years) with autism spectrum disorders (ASD) was published by Kang et al. (2017) [31]. This eight-week exploratory open-label clinical study evaluated the impact of FMT on gastrointestinal tract symptoms (GI) and ASD symptoms [8]. Before FMT, the participants were prepared for the procedure by vancomycin treatment, followed by Moviprep administration. In the study, children with ASD underwent FMT, consisting of an initial multi-stage bowel cleansing, followed by daily administration of a standardized human gut microbiota used in recurrent *Clostridium difficile* infections [43]. The 10-week MTT treatment was followed by an 8-week observation period. Rectal vs. oral initial administration routes were compared for the high initial dose followed by the low oral maintenance dose [31]. After FMT, the patients were assessed with both parent and clinician related scales for ASD symptoms: the Autism Diagnostic Interview–Revised (ADI-R), the Parent Global Impressions-III (PGI-III), the Childhood Autism Rating Scale (CARS), the Aberrant Behavior Checklist (ABC), the Social Responsiveness Scale™ (SRS), and the Vineland Adaptive Behavior Scale, 2nd Edition (VABS-II). The behavioral symptoms of ASD assessed with PGI-II improved significantly, with a sustained follow-up effect at 8 weeks after completion of the treatment [31]. A VABS-II assessment of adaptive behaviors improved for 1.4 years across all domains; however, scores remained lower than their actual age. The GI assessed with the Gastrointestinal Symptom Rating Scale (GRSR) and the daily stool record (DSR) improved for abdominal pain, indigestion, diarrhea, and constipation in a majority of the participants, and remained improved at follow-up. A total of 16 of the 18 study participants reached more than 50% improvement on the GRSR scale [31]. There was no difference between the initial treatment administration route. Only temporary adverse events (AEs) were reported.

The similar design of an open label study was conducted in a following study by Li et al. (2021) [35]. A total of 40 children with ASD accompanied by GI symptoms were treated with FMT for 4 weeks, and followed up for the next 8 weeks. FMT alleviated the ASD symptoms, although most of the scales showed a reversion tendency after 8 weeks. The ABC scale did not show a clear tendency to reversion. The CARS, applied for the assessment of core ASD symptoms, was decreased by 10% at the end of the treatment, and remained decreased by 6% after 8 weeks [35]. The results obtained by the Self-Rating Anxiety Scale (SAS) and SRS were reversed after 8 to 12 weeks without therapy [35]. The average GSRS scores of the ASD children decreased by 35% after 4 weeks of the FMT treatment and, similar to Kang et al. [31], persisted for 8 weeks of follow-up time after finishing the therapy [35]. Similarly, the SRS scores, assessing social skills in ASD children, improved. The parents’ anxiety, assessed with the SAS, decreased with the improvement of ASD-related symptoms and GI scales, but returned to baseline shortly after stopping the therapy. The ABC scale was not reported to reverse after stopping the treatment, while the SRS scale returned to baseline after treatment cessation when assessed on follow-up [35].

Pan et al. (2022) [36] conducted a retrospective cohort study that included 42 children. Their mean age was 6 years, and all children were diagnosed with autism spectrum disorder. Washed microbiota transplantation (WMT) rather than traditional FMT was used. WMT was administered through a transendoscopic enteral tube (TET) at the dose of 60–90 mL/day. The children received up to five WMT courses, each lasting six days. In addition to WMT, patients received other forms of therapy throughout the experiment. The mental health outcomes were assessed using the ABC, the CARS, and the Sleep Disturbance Scale for Children (SDSC). In the ABC assessment, having more WMT courses led to significantly lower ABC scores up to the third WMT course (*p* < 0.05). The scores were significantly lower after each course when compared to the baseline (first, second, and third course: *p* < 0.001; fourth: *p* < 0.01; fifth: *p* < 0.05) [36]. The CARS evaluation showed a decrease in scores as the number of WMT courses increased, but it was insignificant. Nevertheless, for each WMT course, the post-FMT score was significantly improved compared to baseline (first and second course: *p* < 0.0001; third: *p* < 0.001; fourth: *p* < 0.01; fifth: *p* < 0.05) [36]. The SDSC assessment revealed that more WMT courses led to significantly lower SDSC scores up to and including the third WMT course (second course: *p* < 0.05; third course: *p* < 0.01) [36]. The scores were significantly improved after every course, except for the second course (first course: *p* < 0.05, third and fourth: *p* < 0.001; fifth: *p* < 0.05) [36]. The additional effect of WMT was a reduced inflammatory state indicated by globulin and white blood cell levels in serum after the fourth session. The number of white blood cells was positively correlated with the CARS result [36].

In a similar retrospective cohort study, Zhang et al. (2022) [37] analyzed data among 49 children with autism who were suffering from moderate to severe GI disturbances and received WMT. Based on the GI symptoms, participants were divided into two groups: the constipation group (*n* = 24) and the control group (*n* = 25). The assessment was conducted every 2 weeks from the 8th week before WMT to the 8th week after WMT, which was delivered via TET or nasojejunal tube daily (NJT) for 6 days. The participants received two treatments that were administered 4 weeks apart. The second WMT could significantly improve the sleep disorder scores in the constipation group (*p* = 0.026), and the decrease in the SDSC score was synchronized with the increase in the BSFS score; while, in the control group, no significant improvement in sleep quality was noted [37].

A single-arm prospective study by Li et al. (2024) [38] included 98 children with ASD. Their median age was 7 years, and there were 80 males and 18 females in the study. FMT material was obtained from three healthy donors, and the children were divided into three groups based on their tolerance to the capsule FMT treatment. The capsule group (*n* = 73) received a course of FMT capsules for 3 consecutive days, repeated every 4 weeks for a total of three courses. TET administration (*n* = 13) or NJT administration (*n* = 12) was performed for children unable to tolerate capsules. For both groups, one course of fecal solution was infused over 5 min daily, for a total of three courses over 4 weeks. The main outcomes included CARS, ABC, SRS, SDSC, and GSRS. In the capsule group, the outcome assessments were performed at baseline, week 12, and week 20 post-FMT; whereas, for the TET and NJT groups, assessments were conducted at baseline, week 4, and week 12. In the capsule group, there was a significant improvement at 12 and 20 weeks post-FMT in the ABC (*p* < 0.0001), the CARS (*p* < 0.0001), and the SRS (*p* < 0.0001) scores [38]. In the TET group, the ABC noted significant improvement at 12 weeks post-FMT (*p* < 0.05), the CARS noted no significant improvement at either time point, and the SRS showed significant improvement only at 12 weeks post-FMT (*p* < 0.05) [38]. In the NJT group, there was significant improvement in the ABC, CARS, and SRS at both 4 and 12 weeks post-FMT (*p* < 0.01 for all three outcomes). The GSRS scores (median IQR) decreased from the baseline range during the follow-up period [38]. The capsule group and the NJT group showed a decrease in their median SDSC scores from baseline to follow-up time points. The NJT group had a greater reduction in their SDSC scores compared to the other two groups in the between-group analysis [38]. Hazan et al. (2024) [40] provided the first case report on FMT in ASD from a familiar donor in the case of a 19-year-old male with severe autism. The patient was restless with movement stereotypes, making frequent vocalizations, but without verbal contact. He was unresponsive to standard treatments. After the FMT treatment from a typically developing sister, the Autism Treatment Evaluation Checklist (ATEC) of speech increased from 7 to 16.3, and the ATEC of sensory awareness increased from 9 to 24 [40]. A month after the intervention, the patient said his first two words, and his self-injurious behavior stopped. Moreover, the GI symptoms improved [40].

Huang et al. (2022) [41] reported the case of an 18-year-old male patient residing in China who suffered from Asperger’s syndrome and irritable bowel syndrome with diarrhea (IBS-D). His Hamilton Anxiety Scale (HAMA) improved at 1 week (3) and 1 month post-FMT (2) compared to baseline (13), and his Hamilton Depression Scale (HAMD) noted an improvement at 1-week (11) and 1-month (9) post-FMT compared to baseline (15) [41]. Further, his Symptom Checklist-90 (SCL-90) improved at 1 week (242) and 1 month (232) post-FMT compared to baseline (311) [41]. Additionally, the patient reported positive changes in his behavior (reduced auditory hallucinations and delusions, feeling less lost compared to baseline) [41]. After 3 months post-FMT, the patient’s symptoms reappeared, his HAMA score deteriorated but stayed lower than the baseline, and his HAMD and SCL-90 scores deteriorated more compared to the baseline. However, the patient experienced a significant life event at 3 months post-FMT, which might have affected the results [41].

Hu et al. (2023) [42] reported a 7-year-old girl who initially showed typical language development but began to regress at the age of 2 years and 6 months, losing interest in communication and social interaction. By 2 years and 10 months, her Autism Behavior Checklist (ABC) score was 71. At age 4, she became nonverbal, avoided peer interactions, and exhibited stereotypical behaviors. She underwent FMT from a healthy donor. At the end of all five transplant rounds, her scores for the CARS, ATEC, and SRS showed a decreased trend [42], and the overall ABC score fluctuated with a slight downward trend. The number of positive CHAT-23 items significantly reduced after just one round of treatment. The child started saying a single word (like “mom” or “no”), and by the end of the treatment, simple sentences, such as “I love you” [42]. Overall, the data indicated improvement in the child’s social behavior following the FMT treatment.

Recently, a randomized, double-blind, placebo-controlled trial questioned the positive results obtained in smaller open-label trials and case reports. A total of 103 children with ASD from China were randomized, 52 to the FMT group, and 51 to the placebo group [39]. The treatment was performed at weeks 1 and 5, and the follow-up at weeks 9 and 17. The children were assessed with SRS-2 T at week 9 and at week 17 for the primary outcome. The Vineland-3 and ABC scores were assessed for the secondary outcome. There was no difference between the groups for the SRS-2 T, Vineland-3, and ABC scores at baseline, at week 9, and at week 17 [39]. The only significant result differentiating the groups was a statistically significant difference in the socialization domain from baseline to week 17 in the Vineland-3 scale between the FMT and the placebo groups. A significant placebo effect was reported [39]. Although the study has some technical limitations, and the protocol has not been published, it reviews the earlier studies in a critical light.

## 5. Impact of FMT on Gut Microbiota Diversity and Metabolites

As the FMT mechanism lies in gut microbiota modulation and the shift of gut-origin metabolites production, it is important to verify the effect of FMT on gut microbiota composition and metabolism in the selected studies. The results for gut microbiota changes are described in Table 2.

In the baseline of the Kang et al. (2017) trial [31], the children who were diagnosed with ASD had significantly less diverse gut microbiota compared to neurotypical controls. When the initial FMT was delivered 3 weeks after antibiotic treatment, the possibly restored microorganisms were reduced by vancomycin treatment. The following doses in the 10th and 18th weeks after antibiotic therapy significantly increased bacterial diversity compared to baseline (*p* < 0.05 and *p* = 0.001, respectively) [31]. Interestingly, among taxa that became more abundant were bacteria belonging to the *Bifidobacterium*, *Prevotella*, and *Desulfovibrio* genera, significantly surpassing the abundance in the composition of the control group and even the donor, suggesting that during the FMT therapy, the ASD intestinal environment may favor the growth of specific taxa [31]. In the 18th week, the gut microbiota composition of the ASD children became statistically indistinguishable from controls (*p* = 0.78), and increased its similarity to the donor microbiota. After the last application of FMT, the effect persisted for the next 8 weeks [31]. It is worth mentioning that the study had an open-label design, which may instill a placebo effect or bias. Moreover, in the study, FMT was a part of the combination of different treatments targeting gut microbiota. It should be highlighted that the study used a combination of different gut-targeted treatments, so the true effect of FMT may have been hidden [31].

In a subsequent study by Li et. al. (2021) [35], the gut microbiota of ASD and neurotypical children had similar mean species diversity within the groups (α-diversity); however, they differed significantly compositionally between the groups (β-diversity). In the ASD group, relatively more abundant were the bacteria belonging to the *Christensenellaceae* and *Akkermansiaceae* families, as well as the *Christensenellaceae* R-7 group, *Akkermansia*, *Coprococcus* 2, *Eisenbergiella*, and *Tyzzerella* 3 at the genus level. Respectively, the greater abundance of bacteria from the *Peptostreptococcaceae* (at the family level), and the *Romboutsia, Fusicatenibacter*, and *Eubacterium eligens* groups at the genus level was observed in the neurotypical group [35]. After 4 weeks of the weekly FMT treatment, the α-diversity in both groups remained unchanged; however, the β-diversity (unweighted UniFrac) between ASD and Control and ASD and donors reduced [35]. Interestingly, the in-depth analysis showed a positive correlation between a decrease in *Eubacterium coprostanoligenes* abundance after FMT and a decrease in the GI symptoms. After 8 weeks post-FMT, the distances between groups returned to values similar to baseline [35]. Together with gut microbiota changes, the serum concentrations of serotonin and dopamine diminished after 4 weeks of the FMT treatment, with a parallel increase of γ-aminobutyric acid concentration. The concentration of serotonin was negatively correlated and aminobutyric acid was positively correlated with the BSFS. After 8 weeks post-FMT, the concentrations of these neurotransmitters remained altered; however, there was a tendency to return to the pre-FMT state. The design of the study is open-label, which entails similar limits to the previous study. A big advantage over the previous study, however, is that FMT was not combined with the use of an antibiotic, which allowed for exploring the FMT effect more firmly [35].

In the study by Wan et. al. (2024) [39], over 103 children suffering from ASD were randomly assigned in a double-blind, placebo-controlled manner to receive either the FMT or a placebo treatment. The intervention was conducted without any prior pre-treatment (no antibiotic therapy or bowel cleansing), and was divided into two 6-day sessions, occurring in the first and fifth weeks of the study. Immediately after the intervention, a significant increase in gut microbiota α-diversity was observed in the FMT-treated group compared to the placebo group. These differences gradually diminished over time, becoming non-significant by week 9 and returning to baseline levels by week 17 [39]. In week 9, a significant increase in β-diversity was recorded in the FMT group, which was significantly higher than in the placebo group. However, by week 17, β-diversity within the FMT group regressed to baseline levels, with no statistically significant differences remaining [39].

An interesting study, in terms of the duration of the FMT effect, is a case report in which a single dose of FMT was used [40]. This treatment was carried out on an adolescent with severe ASD and was supported by thorough preparation of the intestinal environment for inoculation. For that purpose, vancomycin (500 mg, 3×/day) was administered for 10 days before the procedure, and FMT itself was preceded by deep colonic lavage. The stool for FMT was donated by his sister, possibly increasing the fit of the microbiota to the host. The feces samples were collected from the patient at the study baseline (before antibiotic therapy) and in the 2nd, 6th, 8th, 11th, and 15th months after FMT. Amazingly, throughout the observation period, α-diversity (expressed in Shannon index) increased, and the composition of the microbiota progressively became more similar to the gut microbiota of his sister-donor [40]. By the 15th month, there was a marked reduction of the abundance of *Proteobacteria*, from 26.42% to 0.99%, and *Lactobacillus animalis* (37.14% to 0.00%), alongside an increase in *Actinobacteria* (0.00% to 3.03%), and *Bifidobacterium* rose (0.00% to 1.56%) [40]. Unfortunately, a major limitation is the lack of a sample taken after antibiotic therapy, before FMT, which would be a much better reference point. This would allow for a more thorough and separate study of the effects of antibiotics and FMT on microbiota composition. Compared to the previous two clinical studies that used next-generation sequencing (NGS), this study employed shotgun metagenomic sequencing, which allows for a more detailed analysis of the microbiome as well as an identification of the changes in metabolic pathways. However, an analysis of the metabolic pathway alterations was not performed [40].

Huang et al. (2022) conducted another case study with adolescents with Asperger syndrome and IBS-D [41]. During the study, the patient received FMT without prior antibiotic treatment. The transplantations were performed three times, with two-day intervals between each procedure. Stool and serum samples were collected immediately before the FMT treatment, as well as at 1 week, 1 month, and 3 months after completing the therapy, and were subjected to gut microbiota analysis (by NGS) and circulating metabolite profiling (untargeted metabolomics), respectively. Significant changes in the gut microbiota composition were observed in the 1st week and 1st month after FMT. However, by the 3rd month, there was a partial loss of microbiota shift [41]. The observed alteration of gut microbiota included an increase in the relative abundance of *Roseburia*, *Bifidobacterium*, *Ruminococcus*, *Flavobacteriales*, *Prevotella*, and *Faecalibacterium*, whereas *Coprococcus*, *Dorea*, *Veillonella*, *Clostridium*, *Haemophilus*, *Streptococcus*, and *Romboutsia* showed a decline. At the species level, the relative prevalence of *Bifidobacterium pseudocatenulatum, Ruminococcus bromii, Roseburia* sp. TF10-5, *Roseburia faecis, Faecalibacterium prausnitzii, Prevotella stercorea*, and *Flavobacteriales bacterium* were higher after FMT, with parallel reduced percentages of bacteria belonging to *Bacteroides coprocola, Romboutsia timonensis, Coprococcus catus, Haemophilus parainfluenzae, Dorea longicatena, Bifidobacterium bifidum*, and *Blautia obeum* [41]. After gut microbiota analysis, the untargeted analysis of serum metabolites was conducted, and then metabolic pathways were studied using the Kyoto Encyclopedia of Genes and Genomes database. Several serum metabolites that increased after FMT, such as riboflavin, fructose 1,6-bisphosphate, uridine diphosphate N-acetylglucosamine (UDP-N-acetylglucosamine), and galantamine, were associated with general metabolic pathways (ko01100), secondary metabolite biosynthesis (ko01110), and microbial metabolism in diverse environments (ko01120), and are linked to short-chain fatty acid (SCFA) production, energy metabolism, and vitamin-related pathways [41].

In the Hu et al. (2023) case report [42], a 7-year-old patient with ASD received FMT following a 14-day course of vancomycin. The FMT procedure was performed in five rounds over three months. An NGS analysis revealed significantly reduced gut microbiota diversity in the patient compared to the donor before the procedure. Following FMT, microbial diversity increased to a level comparable to that of the donor. Specifically, the relative abundance of *Bacteroides* and *Ruminococcus* increased, while *Bifidobacterium*, *Anaerostipes*, *Streptococcus*, and *Faecalibacterium* abundance decreased [42]. The potential increase in SCFA production by the new gut microbiota compositions was estimated with the bioinformatics tool PICRUSt analyzing the number of functional genes in a microbiome [44]. Lastly, a colonoscopy revealed multiple lesions and inflamed tissue in the ileum and rectum before FMT, which were significantly ameliorated after the treatment [42].

## 6. Discussion

The pioneering clinical studies on the use of FMT to alleviate the symptoms of ASD have suggested positive clinical results [31,35]. Moreover, gastrointestinal symptoms, including bloating, abdominal discomfort, diarrhea or constipation, were ameliorated. Interestingly, some of the studies showed or suggested changes in the gut microbiota composition and serum concentration of its metabolites. There were no reports of serious AEs during the follow-up period. Ten minor adverse events, such as vomiting, abdominal pain, nausea, fever and headache were noted.

The effects of FMT in ASD individuals mostly persisted [31,35], although there was a reverse trend observed in some of the scales [35,41]. The changes in bacterial microbiota also showed a reverse tendency, and were significant in some of the studies [31,35]. In studies not related to psychiatric diseases, after FMT, the gut microbiota was maintained at 1 year in healthy people [45], and at 2 years for patients with irritable bowel syndrome [46]. In a long-term follow-up study by Kang et al. (2019) [47], 18 patients showed that GI improvement was maintained and ASD symptoms even improved at 2 years after ending the intervention [47]. In other studies on ASD, most of the clinical outcomes were maintained in the follow-up, with only some of the scales reversed (e.g., SAS and SRS in [35]). The reasons for this, and the strategies to retain the effect, remain to be investigated. A recent study by Wan et al. (2024) [39] was the first randomized placebo-controlled study, and it showed a significant symptom improvement in both groups with no difference, though it seems that there could be a significant placebo effect leading to the bias.

FMT also brings changes in the gut microbiota composition. In the studies cited, a reduction in beta diversity was observed between FMT donors and ASD patients-recipients following treatment [31,35,40], as well as between ASD patients and healthy controls [35], indicating a partial change in the gut microbiota towards a ‘normal’ composition. Furthermore, one study [39] has shown an increase in beta diversity in the ASD group after FMT, whereas no such increase was observed among the ASD participants who received a placebo. In this study, pre-FMT procedures, such as bowel cleansing and antibiotic usage, among others, were not performed. This indicates that FMT alone may introduce changes in the gut microbiota, and each patient may have responded differently to the FMT treatment, forming different gut bacterial compositions. The gut microbiota exhibited phylum-level compositional shifts indicative of a more balanced and health-associated profile. *Firmicutes* showed a beneficial restructuring, with an increase in butyrate-producing *Clostridia* (e.g., *Faecalibacterium*, *Roseburia*) and a decrease in potentially pathogenic genera, such as *Clostridium* and *Streptococcus*, supporting improved gut barrier integrity and anti-inflammatory activity [48]. The relative abundance of *Actinobacteria*, particularly *Bifidobacterium*, increased, consistent with enhanced SCFA production and immunomodulatory function [49]. Conversely, a marked reduction in *Proteobacteria*, including *Haemophilus*, indicated a decline in inflammation-associated taxa and potential endotoxin producers, reflecting the restoration of microbial homeostasis [50] (Figure 2).

Gut microbiota dysbiosis has been hypothesized to be involved in the late onset of autism spectrum disorder, accompanied by gastrointestinal symptoms, including bloating, abdominal discomfort, and changes in bowel habits. Certain bacterial neurometabolites may contribute to “leaky gut” and interact with the nervous system [51]. A metagenomic analysis of the gut bacteria in patients with late-onset autism showed a marked decrease in *Bacteroidetes* and an increase in *Sutterella* spp. [52]. Molecular studies have found a greater number of microorganisms that are impaired in autistic children. Altered levels of species of *Bifidobacterium*, *Lactobacillus*, *Sutterella*, *Prevotella*, *Ruminococcus*, and *Alcaligenaceae* have been associated with autism [52,53,54]. Modulation of the gut microbiome, particularly the gain or loss of specific microbial species and pathways involved in the metabolism of SCFAs, tryptophan, and GABA, may merit further investigation as a potential therapeutic strategy for ASD [55].

To investigate the therapeutic effect of FMT, or to confirm the causal mechanisms underlying the effects of the gut microbiota on host physiology and behavior, transplantation of the gut microbiota into animal models (usually mice and rats) is commonly used. These tests are characterized by high reproducibility due to controlled environmental conditions, and reduced inter-individual variability through the use of inbred strains [56]. However, it should be emphasized that a translation of the results obtained in rodent models to humans is limited. Major limitations in the context of the gut microbiota include the difference in the gastrointestinal tract (rodents have a proportionally larger cecum and colon), diet, and general physiology, and therefore have a different gut microbiota composition [56,57]. Due to the different gut conditions in rodents, even using gnotobiotic models, or with prior removal of the gut microbiota, the transplanted human gut microbiota gradually changes, adapting to the new host [56,58]. Furthermore, animal models do not reflect the diversity in the human population [59].

Several ways of interaction of the metabolites secreted through microbiota with the nervous system have been proposed, including local interaction through neurometabolites, direct interaction with the peripheral and central nervous system (CNS), and immunomodulation. Some of the gut microbiota-derived metabolites seem to have neuromodulatory potential either through local interaction or directly influencing the central nervous system (CNS). SCFAs formed during the bacterial fermentation of indigestible carbohydrates, secondary bile acids, tryptophan metabolites, and neurotransmitters, including gamma-aminobutyric acid (GABA), serotonin (5-HT), catecholamine, histamine, and numerous other metabolites, might interact with the nervous system locally via enterochromaffin and enteroendocrine cells localized in the gut lumen, starting an indirect bottom-up signaling process [9,60]. Propionate also activates the gut gluconeogenesis gene expression via a gut–brain circuit involving the free fatty acid receptor 3 [61].

Additionally, it seems that microbiota can directly affect the CNS via the peripheral system. Studies on germ-free mice have shown that the microglia had an abnormal morphology and impaired hippocampal morphology and function [62]. In a mouse model, Goehler et al. (2005) [63] found that the expression of c-Fos, the indicator of neural activity of the vagal sensory ganglia and solitary neurons, increased over time after *Campylobacter jejuni* administration. The c-Fos elevated expression was followed by a rapid increase in neural activity without an increase in proinflammatory cytokines, indicating that the bacteria can directly influence behavior via the vagus nerve [64]. A subsequent study has also shown that early *Campylobacter* spp. infection affects the vagus nerve-mediated neural circuits, leading to anxiety-like behaviors [65].

The gut microbiota also influences endocrine processes. There is evidence that some bacterial metabolites, such as SCFAs, can affect corticosterone production in the ileum, which may affect hypothalamic–pituitary–adrenal (HPA) axis activity [66]. Studies of germ-free mice have shown a hyperresponsive HPA axis and impaired exploratory behavior [56], and reduced anxiety in a stressful situation [67]. Several pathways are associated with a neuromodulatory effect of microbiota-derived metabolites, such as catecholamine, serotonin, GABA, and histamine pathways. The gut bacteria directly impact the brain within the SCFAs that increase the expression of tyrosine hydroxylase, a pivotal enzyme governing the synthesis of dopamine and norepinephrine, as well as dopamine-β-hydroxylase, a rate-limiting enzyme responsible for converting dopamine into norepinephrine [68,69]. However, evidence is still lacking that catecholamines produced by microorganisms affect the CNS, since dopamine synthesized in the periphery cannot cross the blood–brain barrier (BBB) [68,70]. On the other hand, germ-free mice have shown lower levels of tyrosine (the rate-limiting substrate of norepinephrine and dopamine synthesis) than those of former germ-free mice, implying that gut microbiota elevates dopamine levels in the brain [71]. In a study that compared ex-germ-free and germ-free mice, it was found that catecholamine levels were elevated in the brains of the germ-free mice, but inoculation with gut microbiota modulated the catecholamine levels through the turnover of dopamine and norepinephrine in the brain [72].

Another pathway involved is the GABA-related pathway. This pathway was supported by the animal study, in which treatment with propionic acid diminished the levels of GABA, serotonin, and dopamine in germ-free rats [73]. GABA, a major inhibitory neurotransmitter in the CNS, whose dysfunction is associated with depression, anxiety, autism, and schizophrenia, is efficiently produced by *Lactobacillus brevis* and *Bifidobacterium dentium* in the human gut [74,75]. Although GABA is unlikely to cross the BBB, it modulates the brain function indirectly, through modulating the GABA receptors and exosome-mediated signaling, and by serving as a nutrient which is fine-tuning the gut microbiota [76]. *Lactobacillus rhamnosus* was found to reduce anxiety and depression-related behaviors in mice and to increase the concentration of GABA in the hippocampus [77,78].

The gut microbiota affects the metabolism of tryptophan, a precursor for serotonin and kynurenine synthesis. Reductions in tryptophan levels are recognized to be linked with clinical depression [79]. Germ-free mice have been reported to have elevated serum tryptophan levels and reduced blood serotonin levels compared to normal mice, suggesting that tryptophan hydroxylase expression in the gut might be reduced in germ-free mice [80,81].

Histamine, an immunomodulator and neurotransmitter, is involved in the regulatory processes of important functions, such as wakefulness, cognition, circadian rhythm, and neuroendocrine regulation [82]. Some intestinal microbiota can synthesize histamine [83]. Furthermore, a blockade of the histamine receptor H2 has been revealed to reduce mucus secretion and intensify the impairment of the intestinal barrier, which in turn could contribute to the translocation of bacteria into the intestinal lumen via the circulatory system [84].

In the gut–brain axis, a third key player may be the immune system. In ASD, elevated levels of pro-inflammatory interleukins have been observed both in the gut tissues [42] and within the central nervous system (brain tissue [85] and cerebrospinal fluid [86]). Moreover, increased levels of pro-inflammatory cytokines and white blood cells were also found in the bloodstream, the pathway connecting these central hubs of the axis [36,87]. Inflammation in ASD may originate in the gut, due to the aforementioned pathological increase in intestinal permeability and the translocation of bacteria, their metabolites and bacterial cell components into the bloodstream. Among them, lipopolysaccharides and Gram-negative bacterial endotoxins are able to activate pro-inflammatory cytokines release, such as IL-1β, which in turn induces changes in the brain physiology and behavior via the vagus nerve [88]. Another example is propionic acid, a bacterial metabolite, which induced an inflammatory response in human neural stem cells in vitro [89]. It suggests that gut microbiota dysbiosis may lead to low-level inflammation, which may, in turn, affect the brain development [90]. Inflammation may affect the gut microbiota composition, and inflammation can negatively affect the composition of the intestinal microbiota, thus creating a vicious circle that fuels ASD.

Considering the negative results of the recent RCT [39], and the notable placebo effect reported, it is important to highlight that a significant placebo response occurs across ASD studies [91]. In one study, 19% of patients who showed significant improvement were attributed to the placebo effect [92], and up to 30% of ASD participants in clinical trials may exhibit a placebo response [93]. A meta-analysis found a moderate placebo effect in ASD trials (Hedges’ g = 0.45, 95%, confidence interval: 0.34–0.56, *p* < 0.001) [94]. Factors contributing to this effect include low symptom severity at baseline [93], though some conflicting findings exist [95], greater responsiveness to active treatment, and outcome assessments by clinicians rather than caregivers [94]. Additionally, other FMT studies have also reported high placebo response rates [96]. For example, in a study on irritable bowel syndrome (IBS), the placebo group experienced greater symptom relief than the treated group [97].

The main strength of our study lies in assessing the previous research in the context of the recent negative randomized, placebo-controlled trial [39]. At the same time, our scoping review highlights several limitations of the prior studies, including small sample sizes, single-center designs, open-label or retrospective approaches, absence of placebo groups, heterogenous assessment and delivery times, and a potentially high placebo effect influenced by factors such as patient-reported outcome measures [92]. The largest cohort in the most recent RCT included 103 participants, whereas the 2017 study by Kang et al. [31] recruited only 18 subjects. Another potential bias is that, except for the case reports, all studies were conducted in Eastern Asia. Most studies included heterogeneous participant groups in terms of age, ASD symptoms, and use of laxatives. Despite growing interest in FMT as a potential treatment for ASD, all studies and case reports except the recent RCT were rated as having low or very low strength of evidence according to the GRADE system. The recent RCT, assessed as moderate quality by GRADE, also has limitations, including a small cohort from a single center, unpublished detailed methodology, and no registered study protocol in a public database. Taken together, it is important to help non-experts critically assess these studies in light of the most recent evidence and encourage proper methodology for further studies.

This scoping review is subject to several methodological and design limitations. Firstly, as a scoping review, no meta-analysis was performed. Secondly, the review protocol was not registered or published in accordance with PRISMA guidelines. Furthermore, the inclusion of the literature exclusively from the PubMed database may introduce selection bias. While the search strategy was designed to be comprehensive yet focused, it proved challenging due to the rapid proliferation of recent publications containing relevant keywords. The extensive volume of the retrieved literature necessitated thorough screening, further complicating the process. Additionally, there exists a potential publication bias within the field of FMT research in ASD, as studies reporting positive outcomes are more likely to be published than those with negative or inconclusive results.

## 7. Conclusions

Despite initial evidence that FTM might be an effective support treatment in ASD, current evidence remains inconclusive, especially in light of the recent double-blind randomized negative study. Given the high placebo effect in ASD trials and shortcomings in most of the available study designs, high-quality randomized trials are needed before FMT can be recommended for ASD/ADHD in clinical practice.

## Figures and Tables

**Figure 1 microorganisms-13-01290-f001:**
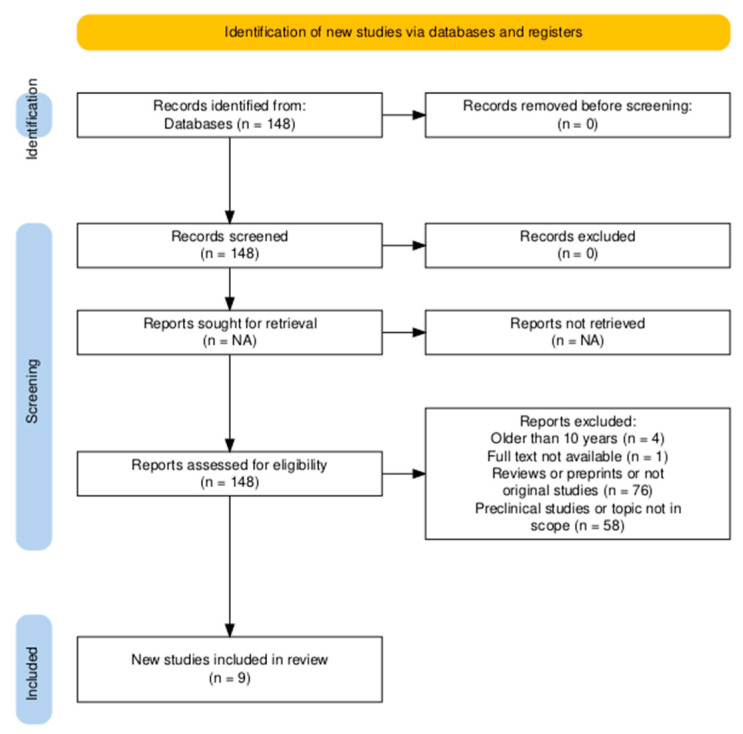
PRISMA workflow.

**Figure 2 microorganisms-13-01290-f002:**
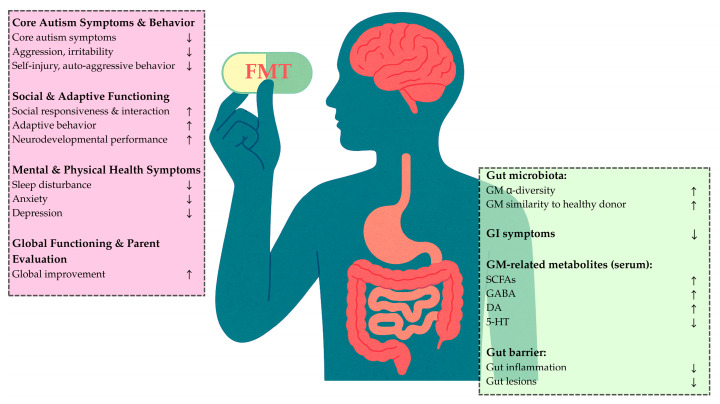
Effect of fecal microbiota transfer on autism spectrum disorders symptoms and gut microbiota composition. Abbreviations: Serotonin (5-HT), Dopamine (DA), Fecal Microbiota Transfer (FMT), Gamma-Aminobutyric Acid (GABA), Gastrointestinal Tract (GI), Gut Microbiota (GM), Short-Chain Fatty Acids (SCFAs). ↑ increased, ↓ decreased.

**Table 1 microorganisms-13-01290-t001:** Clinical outcomes and their persistence in trials investigating the efficacy of fecal microbiota transplantation (FMT) in the treatment of psychiatric diseases.

Study	Type of Study	Intervention	Number of Participants	All Outcome Measures	Outcomes Measures Related to ASD	Observations	Persistence	Strength of Evidence According to GRADE System
Kang et al. (2017) [31]	Open-label	14-day vancomycin + Bowel cleanse + 1 high-dose FMT + 7–8 weeks low-dose FMT	18 (ASD = 10, Control = 8 no intervention on controls)	PGI-III, CARS, SRS, ABC, VABS-II, GSRS, bacterial, and viral changes in stool	CARS	Decreased by 22% from beginning to end of the treatment; *p* < 0.001 (Wilcoxon signed-rank test).	Decreased 24% (relative to baseline) after 8 weeks.	Very low
					PGI-III	Significant Improvement *p* < 0.001 (Wilcoxon signed-rank test).	Maintained after 8 weeks without treatment.	
					SRS	Significant Improvement in social skills; *p* < 0.001 (Wilcoxon signed-rank test),	Maintained after 8 weeks without treatment.	
					ABC	Significant Improvement in multiple behavioral domains (multiple behavioral domains, irritability, hyperactivity, lethargy, stereotypy, aberrant speech); *p* < 0.001 (Wilcoxon signed-rank test).	Maintained after 8 weeks without treatment.	
					VABS-II	Increased by 1.4 years (*p* < 0.001) and across all sub-domain areas.	No data.	
Li et al. (2021) [35]	Open-label	FMT once weekly for 4 weeks (oral capsules or colonoscopy)	56 (ASD = 40, Controls = 16; no intervention on controls)	CARS, ABC, SRS, SAS, GSRS, BSFS, gut microbiota diversity, Serum 5-HT, GABA, and DA	CARS	Decreased by 10% at the end of the treatment and remained decreased by 6% after 8 weeks post-treatment.	Remained decreased by 6% after 8 weeks without treatment.	Low
					SAS	Decreased with the improvement of gastrointestinal symptoms and autism-like symptoms in children.	Returned to baseline after 8 weeks without treatment.	
					SRS	Improved during the treatment.	Reversed after 8 weeks without treatment.	
					ABC	Alleviated by the treatment.	Maintained after 8 weeks without treatment.	
Pan et al. (2022) [36]	Retrospective study	FMT given for 6 consecutive days via TET,	42 ASD (34 males and 8 females)	ABC, CARS. SDSC, BSFS, Rome III criteria, white blood cell (WBC), and globulin levels	CARS	Decreased significantly in comparison to baseline after five/five interventions.	Remained decreased after the fifth intervention.	Very low
					SDSC	Decreased significantly in comparison to baseline after five/five interventions.	Remained decreased after the fifth intervention.	
					ABC	Decreased significantly in comparison to baseline at the third, fourth, and fifth interventions.	Remained decreased after the fifth intervention.	
					Other	Reduced serum level of globulin (third and fourth intervention) and white blood cells (fourth intervention).	Return to baseline levels after the fifth intervention.	
Zhang et al. (2022) [37]	Retrospective study	Single FMT via TET or nasojejunal tube	49 (constipation group *n* = 24, control group *n* = 25, blank group *n* = 24)	CARS, SDSC, ABC, BSFS	CARS	Improved in constipation and control groups.	No follow-up.	Very low
					SDSC	Improved in the constipation group, not improved in the control group.	No follow-up.	
					ABC	Improved in the constipation group, not improved in the control group.	No follow-up.	
Li et al. (2024) [38]	Open-label	FMT once weekly for 4 weeks (oral capsules or colonoscopy)	98 (80 males and 18 females with ASD)	CARS, SRS, ABC, SDSC, GSRS, AEs	CARS	Capsule and NJT and TET groups decreased at the endpoint.	Effect of capsule-based treatment maintained at week 20; effect of NJT and TET maintained at week 12.	Low
					SRS	Capsule and NJT and TET groups decreased at the endpoint.	Effect of capsule-based treatment maintained at week 20; effect of NJT and TET maintained at week 12.	
					ABC	Capsule and NJT and TET groups decreased at the endpoint.	Effect of capsule-based treatment maintained at week 20; effect of NJT and TET maintained at week 12.	
Wan et al. (2024) [39]	Randomized, double-blind placebo	2x 6-day oral FMT capsule administration (in the 1st and 5th weeks)	103 (*n* = 52 FMT group and placebo *n* = 51)	SRS-2 T (primary outcome), Vineland-3, ABC scores. AEs (secondary outcomes)	SRS-2 T	No difference between groups	No difference at week 17 (last follow-up).	Moderate
					ABC	No difference between groups	No difference at week 17 (last follow-up)	
					Vineland-3	At week 9 no difference between groups	Significant difference in the socialization domain from baseline to Week 17 in the Vineland-3	
Hazan et al. (2024) [40]	Case study	10-day vancomycin + Bowel cleanse + single FMT	1	ATEC, CARS, microbiota diversity	CARS	Improvement in verbal communication, behavioral regulation, social interaction, and emotional response, sensory processing and overall functioning.	Maintained at month 16.	Very low
					ATEC	Improvement in speech and sensory awareness.	Maintained at month 16.	
					Other	Decrease in aggression, improvement in sleep patterns and speech.	No data.	
Huang et al. (2022) [41]	Case study	Bowel cleanse + FMT ×3 over 1 week via TET	1	HAMA, HAMD, SCL-90, BSFS, microbiota diversity	HAMA	Strong reduction in overall anxiety symptoms.	Maintained at the 1st month, not maintained at the 3rd month.	Very low
					HAMD	Depression symptoms improved but not fully resolved.	Maintained at the 1st month, not maintained at the 3rd month.	
					SCL-90	A progressive reduction in scores indicates broad improvement in overall psychiatric symptoms.	Maintained at the 1st month, not maintained at the 3rd month.	
Hu et al. (2023) [42]	Case study	14-day vancomycin+ FMT ×5 over 3 months	1	CARS, ATEC, SRS, ABC score, CHAT-23, CNBS-R2016, microbiota diversity, changes in intestinal structure	CARS	Unchanged.	Unchanged.	Very low
					SRS	Overall downward trend after treatment, reflecting improved social abilities.	Trend to decrease at the fifth round of intervention.	
					ABC	Score decreased overall with fluctuations, indicating behavioral improvement.	Fluctuation.	
					ATEC	Slight developmental and physical improvements.	Slight decrease.	
					CNBS-R2016	Progress in multiple areas, including gross motor, adaptive behavior, language, and personal–social. Warning behavior increased.	No data.	
					Other	Speaking single words, staying on the ground.	No data.	

Abbreviations: Aberrant Behavior Checklist (ABC), Autism Spectrum Disorders (ASD), Autism Treatment Evaluation Checklist (ATEC), Bristol Stool Form Scale (BSFS), Childhood Autism Rating Scale (CARS), Chinese Neuropsychological and Behavioral Scale–Revised 2016 (CNBS-R2016), Fecal Microbiota Transplantation (FMT), Gastrointestinal Symptom Rating Scale (GSRS), Hamilton Anxiety Rating Scale (HAMA), Hamilton Depression Rating Scale (HAMD), Nasojejunal Tube (NJT), The Parent Global Impressions-III (PGI-II), Social Adjustment Scale (SAS), Social Responsiveness Scale (SRS), Sleep Disturbance Scale for Children (SDSC), Symptom Checklist-90 (SCL-90), Transendoscopic Enteral Tube (TET), Vineland Adaptive Behavior Scales, Second Edition (VABS-II).

**Table 2 microorganisms-13-01290-t002:** The effects of FMT on the gut microbiota of patients with ASD.

Study	Number of Patients	Gut microbiota Changes	Additional Findings	Follow-Up Duration
Kang et al. (2017) [31]	18 children with ASD (aged 7–16 years)	α-diversity ↑ donor similarity ↑ *Bifidobacterium* ↑ *Prevotella* ↑ *Desulfovibrio* ↑	Donor virome similarity ↑	Bacterial gut microbiota changes are maintained for at least 8 weeks post-treatment.
Li et. al. (2021) [35]	40 children with ASD (aged 3–17 years) + 16 controls	donor similarity ↑ *Eubacterium* *coprostanoligenes* ↓	Comparable effects independent of FMT delivery route. Serum neurotransmitters: 5-HT ↓ GABA ↓ DA ↑ Higher Bristol score correlated with GABA *↑* and 5-HT. ↓ *E. coprostanoligenes* abundance correlated with GABA ↓and GI symptom severity (GSRS) *↑*	The differences between donor and recipient returned to baseline 8 weeks after FMT. Neurotransmitter levels partially reverted toward baseline with a trend of further decline after 8 weeks.
Wan et al. (2024) [39]	103 children with ASD (*n* = 52 FMT group and placebo *n* = 51)	α-diversity ↑	Not reported	A gradual regression of α-diversity to baseline levels was observed by weeks 9 and 17. Increase of β-diversity in the FMT group at week 9, with partial regression towards baseline in week 17.
Hazan et al. (2024) [40]	Case study (aged 19 years with ASD)	α-diversity ↑ donor similarity ↑ *Proteobacteria ↓* *Lactobacillus animalis ↓* *Actinobacteria ↑* *Bifidobacterium ↑*	Not reported	Gradual microbiota improvement throughout 15-month observation, with no signs of remission.
Huang et al. (2022) [41]	Case study (aged 18 years with Asperger syndrome and IBS-D)	Genus level: *Roseburia ↑* *Bifidobacterium ↑* *Ruminococcus ↑* *Flavobacteriales ↑* *Prevotella ↑* *Faecalibacterium ↑* *Coprococcus ↓* *Dorea ↓* *Veillonella ↓* *Clostridium ↓* *Haemophilus ↓* *Streptococcus ↓* *Romboutsia ↓* Species level: *Bifidobacterium* *pseudocatenulatum ↑* *Ruminococcus bromii ↑* *Roseburia* sp. TF10-5 ↑ *Roseburia faecis* ↑ *Faecalibacterium prausnitzii* ↑ *Prevotella stercorea* ↑ *Flavobacteriales* *bacterium* ↑ *Bacteroides coprocola ↓* *Romboutsia timonensis ↓* *Coprococcus catus ↓* *Haemophilus* *parainfluenzae ↓* *Dorea longicatena ↓* *Bifidobacterium * *bifidum ↓* *Blautia obeum ↓*	Altered serum metabolite profile (17 ↑, 37 ↓): Linked to energy, vitamins, and SCFAs metabolism (e.g., riboflavin, UDP-GlcNAc, galantamine) ↑ Involved in amino acid and neurotransmitter biosynthesis (e.g., glutamine, proline, tryptophan) ↓	Microbiota and serum metabolite profiles improved at the 1st month, followed by partial regression at the 3rd month after FMT.
Hu et al. (2023) [42]	Case study (aged 7 years with ASD)	α-diversity ↑ *Bacteroides* ↑ *Ruminococcus* *Bifidobacterium* ↑ *Anaerostipes*↓ *Streptococcus* ↓ *Faecalibacterium* ↓	SCFAs production (predicted) ↑ Inflammation and lesions in the ileum and rectum ↓	Not reported.

Abbreviations: Serotonin (5-HT), Autism Spectrum Disorders (ASD), Dopamine (DA), Fecal Microbiota Transplantation (FMT), Gamma-Aminobutyric Acid (GABA), Gastrointestinal Symptom Rating Scale (GSRS), Irritable Bowel Syndrome with Diarrhea (IBS-D), Short-Chain Fatty Acids (SCFAs), Uridine Diphosphate N-Acetylglucosamine (UDP-GlcNAc). ↑ increased, ↓ decreased

## Data Availability

No new data were created or analyzed in this study.

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
