# Peer review of "The Effects of Fecal Microbial Transplantation on the Symptoms in Autism Spectrum Disorder, Gut Microbiota and Metabolites: A Scoping Review"

_microorganisms, 2025, doi:10.3390/microorganisms13061290_

Round 1

Reviewer 1 Report

Comments and Suggestions for Authors

This scoping review provides valuable insights into FMT's effects on ASD symptoms and gut microbiota, but there are still the following issues that need to be revised:

1.The title mentions "attention deficit hyperactivity disorder (ADHD)", but the paper mainly focuses on ASD and anxiety, with limited content on ADHD. It is recommended to either clarify the findings related to ADHD or adjust the title accordingly.

2.It is suggested to add a "Methods" subsection after "Article Search" (Line 111) to detail the literature screening criteria, data extraction methods, and quality assessment.

3.The titles of Table 1 and Table 2 are nearly identical, which may cause confusion. It is recommended to clearly differentiate them.

4.The discussion should compare the differences between animal models and human studies, highlighting the species-specific limitations of microbiota transplantation.

5.Some references appear outdated. It is advisable to include more recent literature where possible.

6.The journal names in Reference 3 (Line 539), Reference 14 (Line 567), Reference 27 (Line 599), and Reference 36 (Line 626) may not be properly abbreviated. Please carefully check the formatting of all references.

Author Response

Thank you very much for taking the time to review this manuscript. Please find the detailed responses below and the corresponding revisions in track changes in the re-submitted files.

Comments 1: The title mentions "attention deficit hyperactivity disorder (ADHD)", but the paper mainly focuses on ASD and anxiety, with limited content on ADHD. It is recommended to either clarify the findings related to ADHD or adjust the title accordingly.

We changed the title to "The effects of the fecal microbial transplantation on the symptoms in autism spectrum disorder, gut microbiota and metabolites: a scoping review" to better illustrate the main focus of the manuscript. 

Comments 2: It is suggested to add a "Methods" subsection after "Article Search" (Line 111) to detail the literature screening criteria, data extraction methods, and quality assessment.

Response 2: We agree with the Reviewer. The section was added and the Methods section has been edited extensively.

Comments 3: The titles of Table 1 and Table 2 are nearly identical, which may cause confusion. It is recommended to clearly differentiate them.

Response 3: We edited the titles of the tables to as follows:

Table 1. Clinical outcomes and their persistence in trials investigating the efficacy of FMT in the treatment of psychiatric diseases

and 

Table 2. The effects of FMT on the gut microbiota of patients with ASD

Comments 4: The discussion should compare the differences between animal models and human studies, highlighting the species-specific limitations of microbiota transplantation.

Response 4: We added the following paragraph to the disussion:

To investigate the therapeutic effect of FMT or to confirm the causal mechanisms underlying the effects of the gut microbiota on host physiology and behavior, transplantation of the gut microbiota into animal models (usually mice and rats) is commonly used. These tests are characterized by high reproducibility due to controlled environmental conditions and reduced inter-individual variability reduces inter-individual variability through the use of inbred strains [54]. However, it should be emphasized that the translation of results obtained in rodent models to humans is limited. Major limitations in the context of the gut microbiota include the difference in gastrointestinal tract (rodents have a proportionally larger cecum and colon), diet, general physiology, and therefore have a different gut microbiota composition [54,55]. Due to the different gut conditions in rodents, even using gnotobiotic models or with prior removal of the gut microbiota, the transplanted human gut microbiota gradually changes, adapting to the new host [54,56]. Furthermore, animal models do not reflect the diversity in the human population [57]. 

Comments 5: Some references appear outdated. It is advisable to include more recent literature where possible.

Response 5: We agree with the reviewer and have updated the references accordingly. Additionally, during the review process, it was suggested that we include some older foundational research, which we have now incorporated.

Comments 6: The journal names in Reference 3 (Line 539), Reference 14 (Line 567), Reference 27 (Line 599), and Reference 36 (Line 626) may not be properly abbreviated. Please carefully check the formatting of all references.

Response 6: We thank the reviewer for this comment and have edited the references accordingly.

Reviewer 2 Report

Comments and Suggestions for Authors

The review supplied a good idea about relationship about FMT and ASD(or ADHD etc.), and mentioned that the microbiome-gut-brain axis could provide new targets for the prevention and treatment of psychiatric disorders, which could supply a new method for curing illnesses. There are a few suggestions as follows,

L152, Part 3 need to deeply state the cited references, not just include the results of the references.

Table 1, many rensutls of the data were not very apparent, like decreased? I think more detailed data need here. Please check.

Table 2 showed many changes about microbiota, which genus could be the mainly dominant one or more in the related diseases?

Reference [30], SCFAs production increased, please show what kind of SCFAs increased, for there are many kinds of SCFAs.

L276, redundant punctuation;

L366, noname? What is for?

Figure1, what is the trend for beta-diversity?

References: wrong format for the abbreviations of the name of some journals. Please check.

Author Response

Thank you very much for taking the time to review this manuscript. Please find the detailed responses below and the corresponding revisions in track changes in the re-submitted files.

Comments 1: L152, Part 3 need to deeply state the cited references, not just include the results of the references.

Response 1: We elaborated on the results of the study of Kang et al. 2017 and added an assessment of the study in the disussion part.

The description of the study of Kang et al. 2017 in the section 4. Impact of FMT on ASD symptoms reads now as follows:

One of the first studies of FMT on 18 young patients (7-16 years) with autism spectrum disorders (ASD) was published by Kang et al (2017) [31]. This eight-week exploratory open-label clinical study evaluated the impact of FMT on gastrointestinal tract symptoms (GI) and ASD symptoms [8]. Before the FMT, the participants were prepared for the procedure by vancomycin treatment followed by Moviprep administration. In the study, children wit ASD underwent FMT, consisting of an initial multi-stage bowel cleansing followed by daily administration of a standardized human gut microbiota used in recurrent Clostridium difficile infections [41]. 10 weeks MTT treatment was followed by an 8 weeks observation period. Rectal vs. oral initial administration routes were compared for the high initial dose followed by oral low maintenance dose [31]. After FMT, the patients were assessed with both parent and clinician related scales for ASD symptoms: ADI-R (Autism Diagnostic Interview–Revised), Parent Global Impressions-III (PGI-III), Childhood Autism Rating Scale (CARS), Aberrant Behavior Checklist (ABC), Social Responsiveness Scale (SRS), Vineland Adaptive Behavior Scale II (VABS-II). Behavioral symptoms of ASD assessed with PGI-II improved significantly, with a sustained follow-up effect at 8 weeks after completion of treatment [31]. VABS-II assessing adaptive behaviors improved of 1,4 year across all domains, however remained lower than their actual age.  The GI assessed with the Gastrointestinal Symptom Rating Scale (GRSR) and daily stool record (DSR) improved for abdominal pain, indigestion, diarrhea, and constipation in majority of participants and remained improved at follow-up. 16/18 study participants reached more than 50% improvement on the GRSR scale [31]. There was no difference between initial treatment administration route. Only temporary adverse events (AEs) were reported.

Comments 2: Table 1, many rensutls of the data were not very apparent, like decreased? I think more detailed data need here. Please check.

Response 2: We agree with the Reviewer and addedd a detailed description of the results to the table.

Comments 3: Table 2 showed many changes about microbiota, which genus could be the mainly dominant one or more in the related diseases?

Response 3: Following FMT, the gut microbiota exhibited phylum-level compositional shifts indicative of a more balanced and health-associated profile. Firmicutes showed a beneficial restructuring, with an increase in butyrate-producing Clostridia (e.g., Faecalibacterium, Roseburia) and a decrease in potentially pathogenic genera such as Clostridium and Streptococcus, supporting improved gut barrier integrity and anti-inflammatory activity. The relative abundance of Actinobacteria, particularly Bifidobacterium, increased, consistent with enhanced SCFA production and immunomodulatory function. Enrichment of Bacteroidetes, including Prevotella and Bacteroides, suggested a microbiota shift toward fiber-fermenting communities linked to healthier metabolic outcomes. Conversely, a marked reduction in Proteobacteria, including Haemophilus, indicated a decline in inflammation-associated taxa and potential endotoxin producers, reflecting restoration of microbial homeostasis.

Comments 4: Reference [30], SCFAs production increased, please show what kind of SCFAs increased, for there are many kinds of SCFAs.

Response 4:  Thank you for your valuable comment. 

In reference [30], the reported increase in SCFA production was inferred using PICRUSt (Phylogenetic Investigation of Communities by Reconstruction of Unobserved States), a bioinformatics tool designed to predict metagenomic functional potential from 16S rRNA sequencing data. The value presented in the original study reflects the overall predicted increase in overall SCFA production, rather than direct quantification or identification of specific SCFA molecules (e.g. acetate, propionate, butyrate).

To improve the clarity of our manuscript, we have now included an explanation of the method used in this study, specifying that the reported increase in SCFA is a predictive functional inference rather than a measurement of individual compounds.

Comments 5:  L276, redundant punctuation;

Response 5: The punctuation was corrected.

Comments 6: L366, noname? What is for?

Response 6: Thank you for pointing this out. The label “Flavobacteriales noname” refers to bacterial sequences that were taxonomically classified only at the order level (Flavobacteriales), but could not be assigned to any known genus with sufficient confidence based on the available reference database used for taxonomic annotation. To avoid misclassification, we retained the “noname” designation to reflect this level of taxonomic uncertainty.

Comments 7: Figure1, what is the trend for beta-diversity?

Response 7: In the studies cited, a reduction in beta-diversity was observed between FMT donors and ASD patients-recipients following treatment ([22, 23, 28]), as well as between ASD patients and healthy controls ([23]), indicating a partial change in the gut microbiota towards a ‘normal’ composition.

Furthermore, a study [27] showed an increase in beta diversity in the ASD group after FMT, whereas no such increase was observed among ASD participants who received a placebo. In this study, pre-FMT procedures such as bowel cleansing, antibiotic usage, etc., were not performed. This indicates that FMT alone may introduce changes in the gut microbiota and each patient may have responded differently to the FMT, forming different gut bacterial compositions.

As the increase in similarity between donor and recipient was the relationship most frequently demonstrated, it was included in Figure 1. However, in line with your comment, I am posting a full explanation of the changes in the discussion.

Comments 8: References: wrong format for the abbreviations of the name of some journals. Please check.

Response 8: We have corrected the abbreviation formatting.

Reviewer 3 Report

Comments and Suggestions for Authors

The article addresses a highly topical and relevant subject: the use of fecal microbiota transplantation (FMT) in managing autism spectrum disorder (ASD) and attention deficit hyperactivity disorder (ADHD), focusing on clinical symptoms, microbiota composition, and metabolite profiles. A key strength is its attempt to comprehensively summarize current evidence and its alignment with the growing field of gut-brain axis research, which is of significant translational potential. The authors structure the manuscript logically, from introduction through methodology to discussion, and demonstrate awareness of recent developments.

However, substantial limitations must be addressed before the manuscript can be considered for publication. Firstly, although described as a scoping review, the methodology does not fully meet the rigorous standards of a proper scoping review according to PRISMA-ScR guidelines. The paper lacks a flow diagram (e.g., PRISMA chart), a detailed account of the search strategy, and any assessment of the quality of included studies, which is crucial for transparency even in scoping reviews. The methods section must be revised to describe in detail the study selection steps, the quality assessment tools applied, and include a table summarizing study characteristics (e.g., sample size, intervention, outcome measures).

Another concern relates to the statistical interpretation. While the authors present outcomes from various studies, they do not discuss heterogeneity across studies or appraise the strength of the evidence (e.g., using GRADE or similar systems). Including at least a general assessment of the strength of the evidence is imperative, clearly distinguishing between findings from randomized trials and those based on case reports or open-label studies. Additionally, a dedicated section on the limitations of the current evidence should be added, emphasizing the high risk of bias (e.g., placebo effect), small sample sizes, and open-label designs that dominate the current literature.

The discussion section should more explicitly state the limitations of the review itself, including the lack of meta-analysis, possible publication bias, and the restriction to a single database (PubMed), which could impact comprehensiveness. The paper’s strength lies in its critical mention of the latest randomized controlled trial from 2024, which challenges earlier positive results; this demonstrates a commendable critical approach.

The conclusions are notable inconsistencies. While the article concludes that FMT "might be an effective supportive treatment," the presented evidence—especially the negative findings of the randomized controlled trial—does not unequivocally support this assertion. I recommend revising the conclusions to align more closely with the results, clearly stating that current evidence remains inconclusive and that high-quality randomized trials are needed before FMT can be recommended for ASD/ADHD in clinical practice.

Furthermore, I suggest expanding the references. While the manuscript cites recent studies, it would benefit from integrating foundational reviews on the gut-brain axis and consensus guidelines on FMT methodology (e.g., European Society of Gastroenterology or Infectious Diseases Society of America guidelines). Additionally, it would be valuable to include critical reviews discussing the placebo effect in ASD research, as this is particularly pertinent to the topic.

In summary, I recommend deferring the publication decision until the following revisions are made: a fully detailed methods section following PRISMA-ScR standards, inclusion of evidence quality assessment, expanded discussion of limitations and strengths, critical revision of conclusions to reflect the evidence, and enrichment of the reference list. Only after these improvements will the manuscript meet the publication standards of a medical journal and align with EBM principles.

Author Response

Thank you very much for taking the time to review this manuscript. Please find the detailed responses below and the corresponding revisions in track changes in the re-submitted files. 

Comments 1:  Firstly, although described as a scoping review, the methodology does not fully meet the rigorous standards of a proper scoping review according to PRISMA-ScR guidelines. The paper lacks a flow diagram (e.g., PRISMA chart), a detailed account of the search strategy, and any assessment of the quality of included studies, which is crucial for transparency even in scoping reviews. The methods section must be revised to describe in detail the study selection steps, the quality assessment tools applied, and include a table summarizing study characteristics (e.g., sample size, intervention, outcome measures).

Response 1: We agree with the Reviewer. We have added the PRISMA chart and divided the edited extensively methodology section dividing  into 'Article Search' and 'Methodology,' following the advice of Reviewer 1. Additionally, we included information regarding the intervention, outcome measures, and quality assessment tools in Table 1.

Comments 2: Another concern relates to the statistical interpretation. While the authors present outcomes from various studies, they do not discuss heterogeneity across studies or appraise the strength of the evidence (e.g., using GRADE or similar systems). Including at least a general assessment of the strength of the evidence is imperative, clearly distinguishing between findings from randomized trials and those based on case reports or open-label studies. Additionally, a dedicated section on the limitations of the current evidence should be added, emphasizing the high risk of bias (e.g., placebo effect), small sample sizes, and open-label designs that dominate the current literature.

Response 2: We thank the Reviewer for these valuable comments. In response, we have added a dedicated section assessing the available studies, including an evaluation of the certainty of evidence according to the GRADE system. The added paragraph reads as follows:

The main strength of our study lies in assessing previous research in the context of the recent negative randomized, placebo-controlled trial [37]. At the same time, our scoping review highlights several limitations of prior studies, including small sample sizes, single-center designs, open-label or retrospective approaches, absence of placebo groups, heterogenous assessment and delivery times, and a potentially high placebo effect influenced by factors such as patient-reported outcome measures [90]. The largest cohort in the most recent RCT included 103 participants, whereas the 2017 study by Kang et al. [31] involved only 18 subjects. Another potential bias is that, except for case reports, all studies were conducted in Eastern Asia. Most studies included heterogeneous participant groups in terms of age, ASD symptoms, and use of laxatives. Despite growing interest in FMT as a potential treatment for ASD, all studies and case reports except the recent RCT were rated as having low or very low strength of evidence according to the GRADE system. The recent RCT, assessed as moderate quality by GRADE, also has limitations, including a small cohort from a single center, unpublished detailed methodology, and no registered study protocol in a public database. Taken together, it is important to help non-experts critically assess these studies in light of the most recent evidence and encourage proper methodology for further studies.

Comments 3: The discussion section should more explicitly state the limitations of the review itself, including the lack of meta-analysis, possible publication bias, and the restriction to a single database (PubMed), which could impact comprehensiveness. The paper’s strength lies in its critical mention of the latest randomized controlled trial from 2024, which challenges earlier positive results; this demonstrates a commendable critical approach.

Response 3: We have elaborated on the section outlining the limitations of our review. The following paragraph has been added:

This scoping review is subject to several methodological and design limitations. Firstly, as a scoping review, no meta-analysis was performed. Secondly, the review protocol was not registered or published in accordance with PRISMA guidelines. Furthermore, the inclusion of literature exclusively from the PubMed database may introduce selection bias. While the search strategy was designed to be comprehensive yet focused, it proved challenging due to the rapid proliferation of recent publications containing relevant keywords. The extensive volume of retrieved literature necessitated thorough screening, further complicating the process. Additionally, there exists a potential publication bias within the field of FMT research in ASD, as studies reporting positive outcomes are more likely to be published than those with negative or inconclusive results.

Comments 4: The conclusions are notable inconsistencies. While the article concludes that FMT "might be an effective supportive treatment," the presented evidence—especially the negative findings of the randomized controlled trial—does not unequivocally support this assertion. I recommend revising the conclusions to align more closely with the results, clearly stating that current evidence remains inconclusive and that high-quality randomized trials are needed before FMT can be recommended for ASD/ADHD in clinical practice.

Response 4: We agree with the Reviewer and edited the conclusions as follows:

Despite initial pieces of evidence that FTM might be an effective support treatment in ASD, current evidence remains inconclusive, especially in light of the recent double-blind randomized negative study. Given a high placebo effect in ASD trials and shortcomings in most of the available study designs, high-quality randomized trials are needed before FMT can be recommended for ASD/ADHD in clinical practice.

Comments 5: Furthermore, I suggest expanding the references. While the manuscript cites recent studies, it would benefit from integrating foundational reviews on the gut-brain axis and consensus guidelines on FMT methodology (e.g., European Society of Gastroenterology or Infectious Diseases Society of America guidelines). Additionally, it would be valuable to include critical reviews discussing the placebo effect in ASD research, as this is particularly pertinent to the topic.

Response 5: We thank the Reviewer for their comment and have added the following paragraphs, along with the respective references:

On the brain-gut axis

An important puzzle in the development of ASD may lie in a disrupted relationship along the brain–gut axis. The concept that the intestinal microflora can affect the brain, including higher cognitive functions, is well established and supported by animal studies [7–9]. Gut can influence brain through a variety of mechanisms, including gut microbiota composition, immunological processes, production of neuroactive metabolites and vagal interaction  [8]. Recent years brought development in understanding the brain-gut axis with central nervous, gastrointestinal, and immune systems bilateral interactions acting as a one organ [10,11]. Not only can the intestinal microbiota influence the brain, but also the opposite, though the brain to enteric signaling, both directly (though regulating mobility and secretion and gut permeability and indirectly, though secretion of the active substances though lamina propria [10,12].

We expanded the consensus guidelines on FMT methodology:

Fecal Microbiota Transplantation (FMT) is the transfer of stool from a healthy donor to a recipient to restore gut microbiome diversity and balance.. Although the first protocol providing the minimum general steps for FMT [26], further studies to better determine terms of route, protocols optimization, donor-recipient pairing and donor optimization and duration of therapy [27,28] . Both frozen and fresh protocols proved to be successful [26]. FMT can be delivered in multiple routes, among them nasogastric/nasojejunal tube, endoscopy, oral capsules and lower gastrointestinal route (LGI) like retention enema, sigmoidoscopy or colonoscopy. Capsule delivery, either traditional or colon-targeted, is the most commonly applied administration route [29]. Recent American Gastroenterological Association (AGA) guidelines recommended FMT to prevent recurrent C. difficile in select patients, but suggested against the clinical use of FMT for inflammatory bowel diseases or irritable bowel syndrome and at the same time recommended European consensus on FMT also advised to use FMT for the treatment of C. difficile in specialized centers, but found no strong evidence for other applications [26]. Although potential use in other indications, such as other gastrointestinal, metabolic and neurological disorders has been suggested [30]. FMT shows several advantages over probiotic supplementation in ASD: it has longer persistence than probiotics [31] and is more feasible considering the food selectivity in ASD children. Additionally, alternative treatment options for ASD are limited.

On the placebo effect in ASD:

Considering the negative results of the recent RCT [37] and the notable placebo effect reported, it is important to highlight that a significant placebo response occurs across ASD studies [89]. In one study, 19% of patients who showed significant improvement were attributed to the placebo effect [90], and up to 30% of ASD participants in clinical trials may exhibit a placebo response [91]. A meta-analysis found a moderate placebo effect in ASD trials (Hedges’ g = 0.45, 95% confidence interval: 0.34–0.56, P < 0.001) [92]. Factors contributing to this effect include low symptom severity at baseline [91], though some conflicting findings exist [93], greater responsiveness to active treatment, and outcome assessments by clinicians rather than caregivers [92]. Additionally, other FMT studies have also reported high placebo response rates [94]. For example, in a study on irritable bowel syndrome (IBS), the placebo group experienced greater symptom relief than the treated group [95].

Round 2

Reviewer 2 Report

Comments and Suggestions for Authors

Thank you for your revision, and I recommend accept.